# Radiative plasma simulations of black hole accretion flow coronae in the hard and soft states

**Joonas Nättilä** [1,2,3] ✉

Stellar-mass black holes in x-ray binary systems are powered by mass transfer from a companion star. The accreted gas forms an accretion disk around the black hole and emits x-ray radiation in two distinct modes: hard and soft state. The origin of the states is unknown. We perform radiative plasma simulations of the electron-positron-photon corona around the inner accretion flow. Our simulations extend previous efforts by self-consistently including all the pre-valent quantum electrodynamic processes. We demonstrate that when the plasma is turbulent, it naturally generates the observed hard-state emission. In addition, we show that when soft x-ray photons irradiate the system—mimicking radiation from an accretion disk—the turbulent plasma transitions into a new equilibrium state that generates the observed soft-state emission. Our findings demonstrate that turbulent motions of magnetized plasma can power black-hole accretion flow coronae and that quantum electrodynamic processes control the underlying state of the plasma.

Accretion flows around stellar-mass black holes are luminous x-ray sources[1]. The observed x-ray emission is thought to originate from the in-falling plasma close to the black hole. Theoretical accretion flow models predict that the in-falling flow has a two-phase structure forming an optically-thick, geometrically-thin accretion disk and an optically-thin, geometrically-thick diluted "corona" on top (or around) the disk[2,3]. Recent x-ray polarization observations of x-ray binaries[4,5] and numerical accretion disk simulations[6] seem to favor such flow structures.

Long-term monitoring of accreting black holes has also revealed that they exhibit (at least) two observationally distinct states (e.g.,[7]): hard state when the accretion rate is low and soft state when the accretion rate is high. Observed black-hole binary systems, such as Cyg X-1[8], can spend months in one state and then, within a timescale of a week, undergo a rapid state transition into the other state[7]. The physics of such bimodal states has long been speculated (e.g.,[9,10]).

In the hard state, the observed x-ray spectrum is characterized by a power-law with a stable photon index $\Gamma_{ph} \approx 1.7$ (between about 1–100 keV), and a steep high-energy cutoff at $\gtrsim 100$ keV[11]. The hard state is, on average, less luminous but more variable[12]. In the soft state, the observed x-ray spectrum is characterized by a prominent black-body component (with a temperature $k_B T \approx 1$ keV, where $k_B$ is the Boltzmann constant) and an extended high-energy tail[11]. The soft state is, on average, more luminous, with a particularly variable high-energy tail. The emission from both spectral states can be accurately modeled with hybrid (i.e., thermal and non-thermal; e.g.,[13]) plasma distributions[14]. How to actively sustain such a distribution is not known.

A plausible energization source that can power the x-ray emission from the corona is the magnetic field advected (and possibly gener-ated) by the in-falling flow. Such a scenario is also implied by the recent magnetohydrodynamical (MHD) simulations of thick disks (e.g.,[15,16]). Rapid energy release mechanisms from the magnetic field include magnetic reconnection (e.g.,[17–20]) and plasma turbulence (e.g.,[21–25]). However, such categorization is somewhat arbitrary since often reconnection drives turbulence (e.g.,[26]) and turbulence drives recon-nection (e.g.,[23]). Recently, plasma turbulence was demonstrated as a viable energization mechanism for the hard-state emission[25]. That

[1]Department of Physics, University of Helsinki, P.O. Box 64, FI-00014 University of Helsinki, Finland. [2]Physics Department and Columbia Astrophysics Laboratory, Columbia University, 538 West 120th Street, New York, NY 10027, USA. [3]Center for Computational Astrophysics, Flatiron Institute, 162 Fifth Avenue, New York, NY 10010, USA. ✉e-mail: joonas.nattila@helsinki.fi

study included realistic Compton scattering between the turbulent plasma and manually injected photons to recover a realistic hard-state x-ray spectrum.

In this work, we demonstrate that magnetized turbulent plasmas can naturally produce the observed x-ray spectra in both−hard and soft−states when all prominent quantum electrodynamic (QED) processes are self-consistently included. Moreover, we show that turbulent plasmas, in general, exhibit two distinct states: optically thick (producing hard-state-like emission) and optically thin states (producing soft-state-like emission). These states are governed by a pair-thermostat mechanism, which stems from a balance between the QED processes.

## Results

### Magnetized accretion flows

We envision turbulent flaring activity in a collisionless accretion disk corona (see Fig. 1[17-19]) with a characteristic flare emission zone size comparable to the black-hole event horizon size $H \approx r_g \equiv GM_\bullet/c^2 \approx 30$ km,

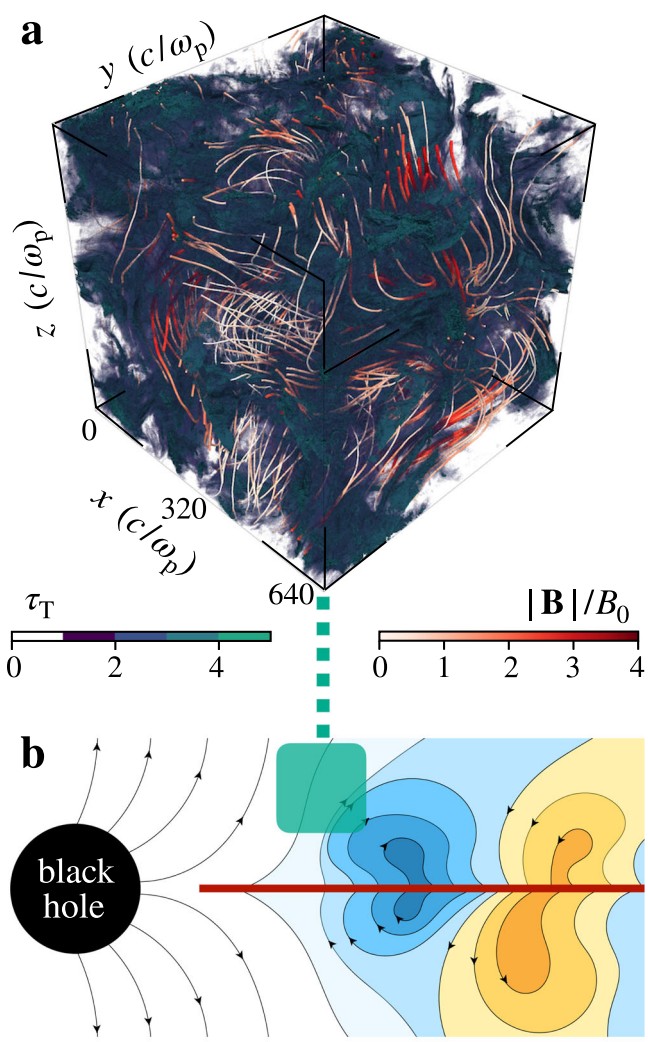

**Fig. 1 | Visualization of the turbulent plasma in the magnetized accretion disk corona. a** 3D rendering of the self-consistent radiative plasma simulation. The plasma (Thomson) optical depth is shown with green isosurfaces at $\tau_T = 1, 2, 3$, and 4. The magnetic field lines **B** are shown with thin tubes. The white-to-red colors denote field strength (in units of the initial guide field $B_0$). **b** Schematic illustration of the accretion flow's magnetic field structures (field strength is shown in blue and yellow colors for clockwise and counter-clockwise polarities, respectively) close to the black hole. The accretion disk is shown with the red horizontal line and the simulated coronal region is highlighted with the green box.

for a black-hole with a mass of $M_\bullet = 20 M_\odot$ (such as Cyg X-1[8]). Here $G$ is the gravitational constant, and $c$ is the speed of light. We assume that the corona is filled with multiple simultaneous (but independent) flaring regions covering the inner (cylindrical) flow region within a radius $r = [10-30]\, r_g$ and a height $H$ comparable to $r_g$. For a flaring efficiency of $\zeta = 0.2$, there are always $N_{flare} = \zeta \pi r^2 H/H^3 \approx 50 - 500$ flaring regions active. The flaring regions have a moderate (Thomson) optical depth $\tau_T$ of about 1 (e.g.,[14]).

We assume that magnetic field lines thread the corona. The magnetic field **B** renders the local plasma magnetically dominated, as quantified by the magnetization parameter

$$\sigma \equiv \frac{U_B}{U_\pm} = \frac{B^2/8\pi}{n_\pm m_e c^2} = O(1), \quad (1)$$

where $U_B \equiv B^2/8\pi$ is the magnetic energy density, $B \equiv |\mathbf{B}| = O(10^7)$ G is the equipartition magnetic field strength (e.g.,[19]), $U_\pm \equiv n_\pm m_e c^2$ is the plasma rest-mass energy density, and $n_\pm$ is the electron-positron number density. Magnetic perturbations propagate with a relativistic Alfvén velocity $v_A \approx c\sqrt{\sigma/(\sigma+1)} \approx c$ in such a medium, and, therefore, introduce a short characteristic timescale of

$$t_A \equiv \frac{H}{v_A} \approx \frac{r_g}{c} \equiv t_g \approx 10^{-4} \text{ s}. \quad (2)$$

Such timescale naturally explains the debated ms-variability observed from accreting systems[12,27,28], since large-scale **B**-field dynamics occur with this timescale.

The magnetic field line motions perturb the corona: magnetic arcades rooted on the disk (or on the black hole) have their footpoints constantly distorted by (Keplerian) shearing motions (e.g.,[6]). As enough twist builds up, the field-line bundle will coil and reconnect with itself, generating strong $\delta B \approx B$ perturbations in the region. The rapidly changing global magnetic field geometry drives strong (relativistic) Alfvénic turbulence (e.g.,[22-24,29]) in the corona. Alternatively, MHD waves supported by the bundle (mostly Alfvén waves) can non-linearly interact and also excite turbulence (e.g.,[30-32]). The resulting injected power density is $\dot{U}_{flare} \approx U_B/t_A$, corresponding to a flare luminosity of $L_{flare} = \dot{U}_{flare}H^3 \approx U_B r_g^3/t_g$ that is about $10^{36}$ erg s$^{-1}$. Then, the accretion flow corona has a total luminosity, as composed of all the ongoing flares, of $L_{corona} = N_{flare}L_{flare}$ that is in the range $10^{37}$-$10^{38}$ erg s$^{-1}$, matching the observed luminosity.

The flare energy density should be compared to a critical limit $U_{crit} \equiv m_e c^2/\sigma_T H$ (i.e., electron rest-mass energy contained in a volume of $\sigma_T H$). For $U_B \gg U_{crit}$, the resulting radiation has enough energy to produce electron-positron pairs and, therefore, alter the flare dynamics. We assume the resulting pair number density is significantly larger than the ion number density, $n_\pm \gg n_i$. The limit can be expressed via luminosity as the so-called (radiative) compactness parameter $\ell_{flare} \equiv L\sigma_T/Hm_e c^3 = O(10)$[33,34]. Values of $\ell_{flare} \gg 1$ imply that pair-creation and other QED reactions are important during the flare. In addition, $\ell_{flare}$ is comparable to $t_g/t_{IC}\gamma$ and $t_g/t_{syn}\gamma$, where $t_{IC}$ and $t_{syn}$ are the Compton and synchrotron cooling times, respectively, of an electron with a Lorentz factor $\gamma$; therefore, $\ell_{flare} > 1$ implies that the radiative cooling times are shorter than the light crossing times in the system. The turbulence must then actively energize the plasma as it is constantly radiating the energy away.

### Hard state

We model the flare evolution with first-principles radiative/QED plasma simulations (see *Methods* section). Our simulation follows the evolution of electron-positron-photon plasma and includes all the prominent QED processes. We include Compton scattering, synchrotron radiation, synchrotron self-absorption, two-photon pair creation, and pair annihilation as simulated processes.

Energy is continuously injected into the computational domain with power $\ell_{inj} = \ell_{flare} = 10$. We observe that the magnetic perturbations injected into the driving scale of $k_0$ quickly forward-cascade and distribute the energy into smaller scales of $k \gg k_0$, indicative of a turbulent cascade. We confirm this picture by constructing the magnetic power spectrum and finding a standard $\propto k^{-q}$ type of scaling with $q \approx 5/3$[35] for the perpendicular wavenumbers. The fully evolved spectra are steeper since radiative drag takes out energy from the cascade[25]. The forward cascade extends to plasma skin-depth scales and steepens after that (e.g.,[22]). The resulting turbulence heats the plasma and energizes non-thermal particles via reconnection to Lorentz factors $\gamma \approx 3\sigma$[22–24]. The stochastic diffusive acceleration, energizing particles beyond $\gamma \gtrsim 3\sigma$, is suppressed by the strong radiative drag[24]. In addition, the Alfvénic turbulence drives trans-relativistic plasma bulk motions with a bulk Lorentz factor $\Gamma \approx$ a few—these bulk motions are crucial for the photon energization[25], as discussed later.

At the beginning of the simulation, energy is removed from the plasma by synchrotron radiation losses. The emitted synchrotron photons have an energy $x_{syn} \approx b\gamma^2$, where $x_{syn} \equiv \hbar\omega_{syn}/m_e c^2$ is the photon energy in units of electron rest-mass, $b \equiv B/B_Q$ is the magnetic field in units of the Schwinger field $B_Q = m_e^2 c^3/\hbar e \approx 4.4 \times 10^{13}$ G[36]. These low-energy photons are, however, almost instantly synchrotron self-absorbed (SSA) by the plasma. SSA competes against Compton scattering in absorbing vs. energizing the low-energy seed photons. Compton scattering rate exceeds SSA rate at $x \gtrsim x_{SSA} \approx 20 x_{syn}$[37] resulting in a characteristic energy of $x_{SSA} = O(10^{-5})$ (0.001 keV) for the synchrotron seed photons.

After the initial transient, the simulation domain is filled with copious synchrotron photons. For our fiducial simulation, we find that the plasma is radiatively dominated with $\eta \equiv U_x/U_\pm = O(10)$, where $U_x = \langle x \rangle n_x m_e c^2$ is the photon energy density and $\langle x \rangle \approx 0.01$ is the mean photon energy. Therefore, the bulk of the energy is stored in the radiation field; however, we emphasize that the plasma is still needed to sustain the electric currents in the system. The seed photons are energized further via Compton scattering with the electron/positron particles—in fact, Compton scattering dominates the radiative losses. In general, Compton cooling rate $\propto U_x$ whereas synchrotron cooling rate $\propto U_B$[36]; we verify that $U_x/U_B = \eta/\sigma > 1$ throughout the simulation. We note that for the simulated parameter regime, even the plasma heating is dominated by the recoil from the Compton scatterings (and to a lesser extent by the turbulent heating); plasma thermalization from SSA remains a secondary process.

The photons are energized via a turbulent bulk Comptonization[19,25]. Here, the turbulent bulk motions drive the photon energization—not the usual thermal motions of the particles. The trans-relativistic MHD motions energize the photons from $x_{SSA}$ to a Wien-like distribution with a peak at $x \approx 0.2$ ($\approx 100$ keV). The process results in a stable photon spectral slope of $\Gamma_{ph} \approx 1.6$ in the energy range between 10 and 100 keV. In addition to turbulent Comptonization, as found by previous simulations[25], we observe a new feedback mechanism where an additional compression of the turbulent plasma (due to the radiative Compton drag) leads to a local increase in the optical depth and further amplification of the photon Comptonization. The optical depth can vary greatly in these intermittent "energization pockets" with $\tau_T$ ranging from 0.1 to 10. In addition, the local scattering environment is modified (see the top panel in Fig. 1): the magnetic field is typically much stronger and less tangled (i.e., $\delta B \lesssim B_0$) inside the optically-thick pockets (with $\tau_T \gtrsim 1$); in contrast, the magnetic field is more tangled (i.e., $\delta B \approx B_0$) outside these regions (with $\tau_T \lesssim 1$). These spatio-temporal turbulent fluctuations self-consistently produce and sustain a hybrid plasma distribution.

The radiation spectrum from the fiducial simulation setup is shown in Fig. 2 (top panel; black curve). It has a striking similarity with the observed hard-state spectra of Cyg X-1: a photon spectral index $\Gamma_{ph} \approx 1.6$, pronounced peak at around 100 keV, and a sharp cutoff at

higher energies. The resulting spectrum is set only by our selection of the flare-region size $H = 30$ km and injection power $\ell_{inj} = 10$; all the other parameters, such as the optical depth $\tau_T \approx 1$ and plasma magnetization $\sigma \approx 1$, emerge self-consistently from the pair creation and annihilation balance. These resulting parameters are similar to those obtained by spectral fitting[14] and when comparing radiative plasma simulations of magnetized turbulence to the hard-state observations[25].

Finally, our simulations demonstrate that the in-situ generated SSA photons can seed the hard state Comptonization—no external photons are needed to recover the correct spectral shape. This implies that in the hard state, the inner accretion flow can structurally resemble the so-called hot accretion flow geometry: no optically thick inner disk component is required, as is typically assumed, e.g., in the "sandwich" disk models (see, e.g.,[38] for a review).

## State transitions

Our self-consistent simulations with a complete set of QED processes demonstrate that x-ray emission resembling the hard state naturally emerges from magnetized, turbulent plasma. However, accreting x-ray binaries are observed to change from the hard state into the soft state when the accretion rate increases (e.g.,[1]). When the mass transfer increases, the inner flow region becomes populated by additional soft x-ray photons[39]. A change in the external photon flux indeed turns out to play a key role in inducing a state-transition-like change in the plasma, as we demonstrate next.

We examine the effect of an ambient photon background—and the emergence of the soft state—with additional simulation, in which we illuminate the numerical domain with external soft x-ray photons that mimic those from the inner disk region. Physically, the inner disk region is expected to be optically thick and to produce soft x-ray photons with an average energy $x_{ext} = O(10^{-3})$ (1 keV)[2]. For simplicity, we approximate the multi-color disk black-body emission with a single Planck spectrum with a temperature $T_{ext} \approx 0.3$ keV and luminosity $\ell_{ext}$. We do not vary $\ell_{ext}$ during the simulations and, therefore, do not model the slow transitions directly (observed transitions last weeks whereas eddy turnover times last less than seconds); instead, we perform multiple independent simulations with different $\ell_{ext}$.

Figure 2 visualizes the escaping radiation spectra from the flare regions with $\ell_{ext}$ increasing from 0 to 10. We find that when the luminosity of the external photons exceeds $\ell_{ext} \gtrsim \ell_{crit} \approx 1$, the simulated medium transitions into another state during about $3t_0$. This transition also helps us explain the origin of the plasma conditions required for the soft-state radiation. In this new state, the average plasma optical depth drops from $\tau_T \approx 1$ to 0.3, and the peak of the escaping radiation shifts from about 100 keV to 1 keV. In general, we find that the system's state depends sensitively on $\ell_{ext}$ exceeding $\ell_{crit}$; physically, the threshold corresponds to feeding more external photons into the system compared to the (up-scattered) SSA-seed photons (see also Supplementary Discussion).

The decrease in the optical depth is driven by an enhanced pair annihilation. First, the external luminosity cools down the plasma (we measure temperatures of $\theta_\pm \equiv k_B T_\pm/m_e c^2 \approx 0.2$ for $\ell_{ext} = 10$, where $T_\pm$ is the plasma temperature; whereas $\theta_\pm \approx 1$ for $\ell_{ext} = 0$). The resulting cold medium has an increased pair annihilation rate that reduces the plasma number density by a factor of $\approx 3$ compared to a test simulation without pair annihilation. However, localized energization regions in the turbulence can still sustain $\gamma \gg 1$ on some parts of the domain and will, therefore, produce a strong nonthermal component to the emission (see Fig. 3). The reduced optical depth and the local nonthermal plasma pockets make the soft x-ray band more pronounced (because some external photons can escape without interacting) and the hard x-ray band more prominent (because nonthermal pairs Compton up-scatter photons into a flat high-energy tail). Overall, the resulting spectrum is strikingly similar to the observed soft-state

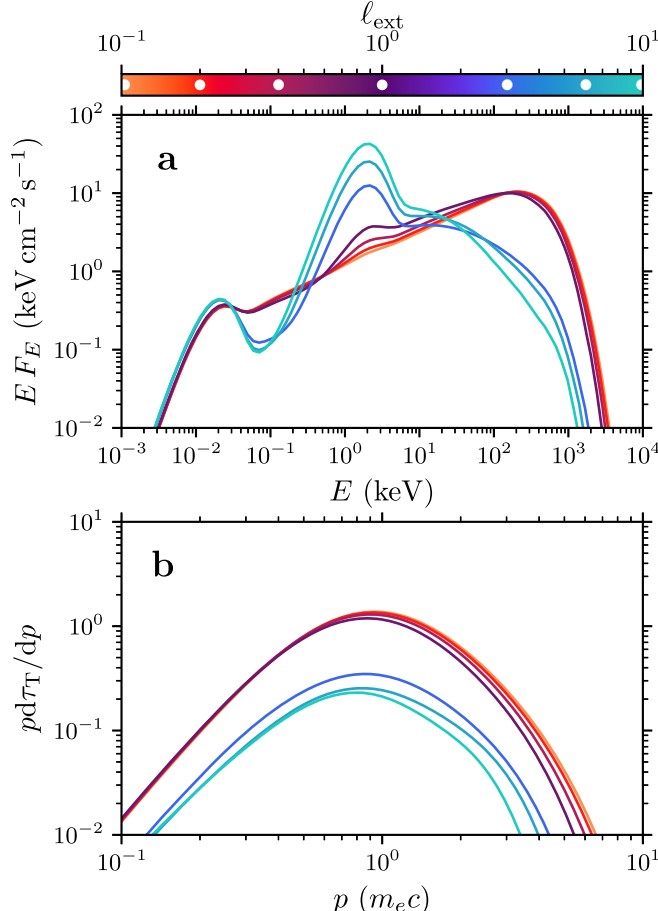

**Fig. 2 | Photon and plasma spectra as a function of the external luminosity $\ell_{ext}$.**
**a** Escaping photon flux for simulations with $\ell_{ext}$ = 0.1, 0.2, 0.4, 1, 3, 6, 10 (orange to turquoise curves). The values of $\ell_{ext}$ are denoted with white dots in the colorbar.
**b** Self-consistent plasma momentum distribution (in units of optical depth $\tau_T$) for the same simulations. The spectra are provided as Source Data files.

radiation of Cyg X-1 and other well-studied XRB systems such as MAXI J1820 + 070[40].

## Discussion

It is natural to assume that the disk coronae are turbulent and magnetically active, similar to the solar corona[17–19]. We have demonstrated that first-principles simulations of turbulent plasmas with a complete set of QED processes can self-consistently reproduce the required plasma conditions and the observed soft and hard state radiation spectra from black-hole x-ray binaries. In Fig. 4, we show the escaping radiation from simulations resembling the hard state (with $\ell_{inj}$ = 10 and $\ell_{ext}$ = 0.4) and soft state (with $\ell_{inj}$ = 10 and $\ell_{ext}$ = 6). The total radiative output is obtained by summing over multiple independent flaring regions, mimicking the total emission from the inner region of the accretion flow. The spectra are attenuated with a photoelectric absorption and modified by Compton reflection from the disk as detailed in the Supplementary Discussion. We compare the resulting theoretical calculations to four Cyg X-1 observations, consisting of two hard-state and two soft-state spectra[7]: the observations are well-described by a magnetized corona that either self-generates the synchrotron seed photons (hard state; low accretion rate) or reprocesses the incoming soft x-ray photons from the disk radiation (soft state; high accretion rate). Interestingly, these simulations reproduce the qualitative spectral shapes and demonstrate similar energy-dependent variability, with the high-energy tail mostly changing. Direct spectral fits and recovery of exact model parameters for the Cyg X-1 and other

XRB systems are postponed to the future since such comparisons would require an extensive model library that is currently computationally unfeasible to produce.

Furthermore, our simulations exhibit distinct changes in the plasma state as a function of the external seed-photon luminosity $\ell_{ext}$. We track such changes by performing otherwise identical simulations with different fixed $\ell_{ext}$; the turbulent plasma becomes optically thin for runs with $\ell_{ext}$ > 1. Such a dependency supports the picture where x-ray binary state transitions can be simply explained by a change in the disk geometry (e.g.,[14]): as the inner disk evolves and provides more seed photons, the corona reprocesses this radiation and relaxes to a specific plasma state that shapes the radiation into the observed spectra. The required plasma parameters are self-consistently set by the thermostatic properties of the pair plasma[41] (see also Supplementary Fig. 1). Intermittent energization by magnetized plasma turbulence, on the other hand, naturally sustains the hybrid plasma distribution in both hard and soft states via the feedback mechanism where the turbulent energization zones become enshrouded by the pair-cascade clouds, resulting in increased (local) photon thermalization. Previous studies have found similar indications for the hard state[25]; here, our fully self-consistent simulations verify this picture for both states by including all the prominent QED processes. The remaining plasma physics question to be explored with next-generation simulations is the possible role of ions; here, we specifically assumed that $n_\pm \gg n_i$ based on the efficient pair creation.

We assume that the corona is filled with multiple (on the order of 50–500) independent flaring regions with a size comparable to $r_g$[17,18]. One interesting consequence of such a picture is that the observed spectra should vary on a short timescale of 0.1 ms; some hints of such variability are indeed seen in historical data[12]. On the other hand, an obvious mismatch between the current simulations and observations is the weaker nonthermal tail at photon energies ≳1 MeV[42]. Such high-energy tails can maybe be produced by more powerful (and less frequent) flares with $\ell_{inj} \gg 10$ (possibly of different origin, such as from the jet). Another interesting future avenue is to test the expected polarization signatures from a turbulent corona. Such results can then be directly compared to the recent IXPE observations[4,5].

## Methods

We simulate the radiative plasma dynamics with the open-source plasma simulation toolkit RUNKO using the publicly available V4.1 RIPE KIWI release[43,44] (github.com/hel-astro-lab/runko; commit FF1B5A6). The plasma dynamics are modeled using the particle-in-cell (PIC) technique, and the QED processes (i.e., interactions between plasma and radiation) are modeled using the Monte Carlo method. Importantly, the simulated photon particles have adaptive weights that are fine-tuned to resolve the QED processes accurately[45].

Our fiducial simulation has a resolution of $640^3$ grid points with a grid-spacing of $\Delta x = c/\omega_p$ (where $\omega_p^2 \equiv 4\pi n_\pm e^2/m_e$ is the plasma frequency), corresponding to a total box size of $l_0 = 640\, c/\omega_p$. The computational domain is triply periodic. The adaptive Monte Carlo algorithm sets the $e^\pm$ particle resolution to ≈ 10 particles per cell and the photons to ≈200 particles per cell. We impose an initial magnetic field of $\mathbf{B}_0 = B_0\hat{\mathbf{z}}$ into the domain (with a strength corresponding to an initial magnetization of $\sigma_0$ = 1). The turbulence is continuously excited in the domain with an oscillating (Langevin) antenna[46]: the antenna drives magnetic perturbations with an amplitude of $\delta B/B_0$ = 0.8 on small wavenumbers $k_0 = 2\pi/l_0$ with a frequency of $\omega_{ant} = 0.8\omega_0$ (where $\omega_0 \equiv 2\pi/t_0$ is the eddy-turnover frequency, and $t_0 \equiv l_0/v_A$ is eddy-turnover time) and decorrelation time of $\omega_{dec} = 0.6\omega_0$. The resulting turbulence dynamics are found to be insensitive to the details of the antenna. The injected energy is removed from the box by the self-consistently created photons: an energy balance of $n_x \langle x_{esc} \rangle/t_{esc} \approx \delta B^2/8\pi t_0$ is reached by $t \approx 4t_0$, where $n_x$ is the photon (number) density, $\langle x_{esc} \rangle = \hbar\omega_{esc}/m_e c^2$ is the mean photon energy of the escaping radiation

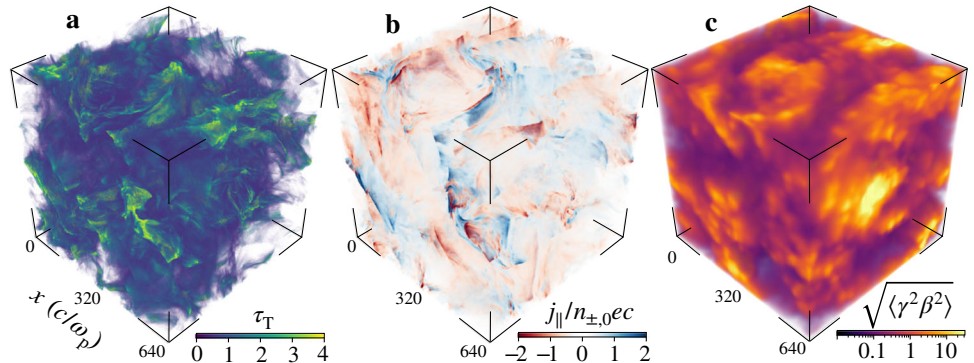

**Fig. 3 | Intermittent fluctuations in the magnetized plasma turbulence.**
**a** Pair-plasma (Thomson) optical depth $\tau_T = n_\pm \sigma_T H$, where $n_\pm$ is the local plasma number density, $H$ is the system size, and $\sigma_T$ is the Thomson cross-section. **b** Parallel current density $j_\parallel / n_{\pm,0} ec$, where $j_\parallel \equiv \mathbf{j} \cdot \mathbf{B}/B$, $\mathbf{j}$ is the local current density, $\mathbf{B}$ is the magnetic field, $e$ is the electron

charge, and $c$ is the speed of light. **c** Proxy of the (local) radiative output power of the plasma $\sqrt{\langle \gamma^2 \beta^2 \rangle}$, where $\beta$ is the flow's coordinate bulk velocity and $\gamma$ is the bulk Lorentz factor. The 3D simulation data outputs are available from[47].

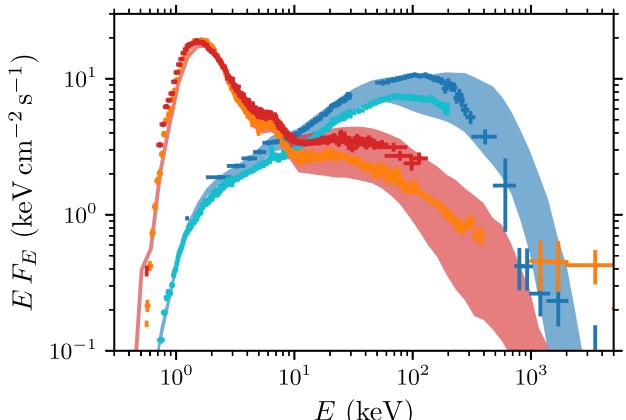

**Fig. 4 | Comparison of the simulated photon spectra to Cyg X-1 observations.**
The bands show the minimum and maximum flux from the simulation during a time window $t/t_0 \in [3, 10]$, where $t_0$ is the eddy turnover time. A simulation output with weak external photon flux (blue band; $\ell_{ext} = 0.4$) is compared to two hard-state observations (blue and cyan crosses; $1\sigma$ error bars). A second simulation with a high external photon supply (with $\ell_{ext} = 6$) is compared to two soft-state observations (red and orange crosses). The x-ray observations are performed by CGRO/BATSE and RXTE/ASM instruments as reported in[7]. The spectra are provided as a Source Data file.

(in units of $m_e c^2$), and $t_{esc} \approx H\tau_T/c$ is the escape time. We remove the computational photons from the domain using an escape probability formalism. All simulations are evolved up to $t/t_0 = 10$. More technical details of the simulations are given in the Supplementary Methods.

## Data availability
The processed data for Figs. 2, 4, and the Supplementary Fig. 1 are provided in the Source Data file. The full 3D simulation data output files for Fig. 1 and 3 are available in Zenodo (https://doi.org/10.5281/zenodo.12743684, https://zenodo.org/records/12743684)[47]. The datasets generated during and/or analyzed during the current study are available from the corresponding author on request. Source data are provided with this paper.

## Code availability
The simulation code is available from github.com/hel-astro-lab/runko, https://zenodo.org/records/10912804[44]. All other relevant data analysis scripts are available from the corresponding author on request.

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

## Acknowledgements

J.N. thanks Daniel Grošelj for useful discussions and comments. This work is supported by ERC grant (ILLUMINATOR, 101114623). Views and opinions expressed are however those of the author(s) only and do not necessarily reflect those of the European Union or the European Research Council. Neither the European Union nor the granting authority can be held responsible for them. Simons Foundation is acknowledged for computational support.

## Author contributions

J.N. developed the simulation code, performed the simulations, analyzed the data, performed the theoretical analysis, and prepared the manuscript.

## Competing interests

The author declares no competing interests.
