## [Peer Review File · Nature Communications]

REVIEWER COMMENTS

Reviewer #1 (Remarks to the Author):

The authors addressed my issues and well revised the manuscript. I read the manuscript again and found several relatively minor issues, which the authors should address before it is published in Nature Communications.

Comments:

1. In P.4, as the origin of perturbation of the magnetic field, the authors consider magnetic arcades rooted on the disk or the black hole. It is better to cite some MHD simulation work.

2. In Fig.2, where is $I_{\text{ext}}=1$ line? It is hard to see from the figure. I feel the state is suddenly changed (blue to red curves), in particular $E \sim 10^{-1}, 10^0, 10^3$ regions. Why so quickly change? If the authors run the simulations with $I_{\text{ext}} \sim 0$ such as 0.001, will the result stay between blue and red curves?

3. In this work, the authors applied the spectra to Cyg X-1. It is possible to apply other X-ray binaries? I think it is worth to comment on the applicability to other sources.

4. In the main text, the author used B-field. I think it is better to use a magnetic field.

5. In the Method section, the authors excited the turbulence in this simulation domain. Is this turbulent excitement at the initial time only? Or do you continuously excite the turbulence in this simulation domain?

6. In the Method section, what is the boundary condition for these simulations?

Reviewer #2 (Remarks to the Author):

I appreciate the author's replies to the report, the changes in the manuscript, and my main questions and criticisms have been carefully handled! I do have one main point left, and two smaller remarks:

main point 1.

The authors argue that the main point is not the (similar) results to Groselj23, but the state transition itself (or rather the two states separately).

However, the state transition is in my opinion the main weak point of the paper. A true simulation of the state transition should include the actual transition, dynamically, without changing external conditions like photon feeding. That would require in my opinion global simulations including the accretion flow dynamics and irradiation of the coronal region. I understand this is prohibitively expensive (even if assuming MHD without all the self-consistent physics of the simulations presented in this work). However, therefore my recommendation remains that this is a very good and robust paper, yet not a groundbreaking result.

If the work focuses on the extreme plasma physics (as mentioned in the decadal), I do agree it is a great advancement. However, the focus on the work on state transitions I still find somewhat problematic due to what I stated above, the phase transition is not dynamically probed, but put in by hand in a sense (by choice of irradiating photons, and assuming a certain state of turbulence for the corona). I understand this is motivated by observational evidence, and is also currently what is possible (and definitely computationally state of the art). But I am not certain a computationally state of the art plasma physics result warrants publication in Nature Communications, and I am not convinced the state transition problem is being solved by these two simulations of the two different states. Are the authors absolutely convinced this problem is now solved, and this is the only possible explanation of hard and soft states, without having looked at the role of ions, and the role of global dynamics forming the turbulence, the radiative processes and the content of the coronal region? I am not convinced, but happy to be persuaded otherwise if the authors can genuinely argue that this problem is now solved.

I do appreciate the rephrasing of the title and wording in the manuscript, however, as the main selling point of the paper being the state transition / the two states, I am still inclined to reject for Nature Communications due to a lack of impact on the topic that the authors mention as the main result.

Smaller point 1.

This is a smaller point, but I would not argue that it is hard to imagine how ions can get mixed in. Multiple papers have shown that there are clear mixing regions between disks, jets, jet boundaries/sheaths and coronal regions recently. I would turn the argument around and instead of "given the author's results, ions are not needed" it should be understood what the contribution of

ions would be. I understand this will not be done for this work, but it is an essential part of solving the coronal problem!

Smaller point 2.

A second smaller point, is the citation of thick magnetically arrested disks (papers by Liska on dynamo in thick disks, and Ripperda on thick disk MADs) in the reply to referee 1: It is completely unclear whether magnetized regions in thick disks are in any way representative for thin, luminous XRB disks.

Smaller point 3.

A third smaller, yet potentially important, point: It is quite unclear how the authors get synchrotron emission. as they do not get any power law, because the effective γ_{rad} is ~ 1 . So that would make me think that the process has to be thermal "cyclotron". However, thermal cyclotron would for real XRB coronal parameters be self absorbed. So it seems that there is some significant fine tuning to stop SSA for thermal photons? Is this potentially because the compactness is rather low, resulting in high electron temperature? This would potentially explain producing more cyclotron photons. I think this is an important point to address, and in my opinion shows that despite the first principles nature of these simulations, the results still depend strongly on assumptions and tweaking of parameters, hence my reservation to declare the two states as a solved problem.

This is our response to the third round of referee comments. We thank the referees for their comments. Our response is given below. All the additions to the manuscript are marked with blue text; all the removals are marked with red & ~~striketrough~~-text.

Reviewer #1 (Remarks to the Author):

The authors addressed my issues and well revised the manuscript. I read the manuscript again and found several relatively minor issues, which the authors should address before it is published in Nature Communications.

Comments:

1. In P.4, as the origin of perturbation of the magnetic field, the authors consider magnetic arcades rooted on the disk or the black hole. It is better to cite some MHD simulation work.

Reply:

Done, we now cite Liska et al 2022 when discussing "footpoints constantly distorted by (Keplerian) shearing motions [e.g., 6]".

2. In Fig.2, where is $I_{\text{ext}}=1$ line? It is hard to see from the figure. I feel the state is suddenly changed (blue to red curves), in particular $E \sim 10^{-1}$, 10^0 , 10^3 regions. Why so quickly change? If the authors run the simulations with $I_{\text{ext}} \sim 0$ such as 0.001, will the result stay between blue and red curves?

Reply:

This is indeed an important point to clarify in the manuscript. We find that the optically thick hard state (with photon peak at ~ 100 keV) and the optically thin soft state (with photon peak at ~ 1 keV) are so-called "attractors" of the underlying chaotic dynamical system. Because of this, the turbulent flow undergoes a sharp transition from one state to the other when the external flux exceeds the critical $I_{\text{ext}} > I_{\text{crit}} \sim 1$ threshold, and the dynamical system changes its preferred attractor. Sect. 4 now emphasizes this fact. In addition, we provide a more in-depth discussion of the mechanism in a new Appendix B.

Finally, we have changed the colormap in Fig 2 and highlighted the selected I_{ext} values in the color bar to help the reader better interpret the figure.

3. In this work, the authors applied the spectra to Cyg X-1. It is possible to apply other X-ray binaries? I think it is worth to comment on the applicability to other sources.

Reply:

Excellent point; we have added a note about the generality of the results to Sect 4 and 5. Specifically, we also highlight the observations of MAXI J1820+070 with comprehensive multiwavelength observations during both soft and hard states (e.g., Srimanta et al. 2024).

4. In the main text, the author used B-field. I think it is better to use a magnetic field.

Reply:

Done; we have changed the occurrences of "B-field" to "magnetic field."

5. In the Method section, the authors excited the turbulence in this simulation domain. Is this turbulent excitement at the initial time only? Or do you continuously excite the turbulence in this simulation domain?

Reply:

We study continuously driven turbulence. We clarified in Sect 3 and the Method section that "Energy is `_continuously_` injected into the system." i.e., the system is driven.

The choice is, however, rather generic because recent first-principles studies support the finding that, if the simulated box size is large enough, there is no qualitative difference between decaying vs. driven turbulence in terms of plasma energization (e.g., Zhdankin et al. 2017 exploring driven turbulence vs. Comisso & Sironi 2018 exploring decaying turbulence". We have now detailed this in Appendix A when discussing the numerical convergence.

6. In the Method section, what is the boundary condition for these simulations?

Reply:

The simulations are triply periodic. The method section now specifies that the "Domain is triply periodic." This is a standard choice in controlled local numerical experiments.

Reviewer #2 (Remarks to the Author):

I appreciate the author's replies to the report, the changes in the manuscript, and my main questions and criticisms have been carefully handled! I do have one main point left, and two smaller remarks:

main point 1.

The authors argue that the main point is not the (similar) results to Groselj23, but the state transition itself (or rather the two states separately). However, the state transition is in my opinion the main weak point of the paper. A true simulation of the state transition should include the actual transition, dynamically, without changing external conditions like photon feeding. That would require in my opinion global simulations including the accretion flow dynamics and irradiation of the coronal region. I understand this is prohibitively expensive (even if assuming MHD without all the self-consistent physics of the simulations presented in this work). However, therefore my recommendation remains that this is a very good and robust paper, yet not a groundbreaking result.

If the work focuses on the extreme plasma physics (as mentioned in the decadal), I do agree it is a great advancement. However, the focus on the work on state transitions I still find somewhat problematic due to what I stated above, the phase transition is not dynamically probed, but put in by hand in a sense (by choice of irradiating photons, and assuming a certain state of turbulence for the corona). I understand this is motivated by observational evidence, and is also currently what is possible (and definitely computationally state of the art). But I am not certain a computationally state of the art plasma physics result warrants publication in Nature Communications, and I am not convinced the state transition problem is being solved by these two simulations of the two different states. Are the authors absolutely convinced this problem is now solved, and this is the only possible explanation of hard and soft states, without having looked at the role of ions, and the role of global dynamics forming the turbulence, the radiative processes and the content of the coronal region? I am not convinced, but happy to be persuaded otherwise if the authors can genuinely argue that this problem is now solved.

I do appreciate the rephrasing of the title and wording in the manuscript, however, as the main selling point of the paper being the state transition / the two states, I am still inclined to reject for Nature Communications due to a lack of impact on the topic that the authors mention as the main result.

Reply:

We strongly believe that the reported work has a high impact because it is:

i) a first-principles proof that magnetized turbulent plasma can explain the hard and soft state observations. This demonstrates to the accretion disk community that turbulence and magnetic field dynamics are crucial for interpreting the data (as speculated already by Galeev et al. 1979);

ii) the first direct demonstration that plasma in the corona has a natural tendency to converge on the required plasma parameters that then self-consistently reproduce the hard and soft state observations. This solves the problem of why the coronal plasma has the parameters it has: it is a natural outcome of injecting turbulent energy into the system; and,

iii) the first fully self-consistent radiative/QED plasma simulation and, as the referee remarked, "a great advancement" in the field of extreme plasma physics.

We do agree with the referee's points that:

i) the role of ions and plasma composition needs to be probed. However, such studies will be more incremental since here the dominant physics is controlled by the QED reactions between electron-positron plasmas and the radiation field (ions can contribute, e.g., via secondary effects such as electron-ion heating),

ii) global disk dynamics need to be simulated in the future. However, such future simulations will still most likely reproduce our reported results where the turbulent diluted coronas on top of the disk will exhibit the reported thermostatic behavior and control the radiation.

iii) more complex radiative processes need to be explored. However, such studies will be incremental since we are the first to self-consistently include all the single and two-body QED processes (further studies can, e.g., explore radiative corrections such as Double Compton that affect the seed photon generation on a ~1-5% level, as estimated from our current plasma parameters).

We have now better emphasized these novelties and findings:

- The abstract now emphasizes our main findings with items *i)* and *ii)*
- The abstract now highlights that the corona is a result of extreme plasma physics processes
- The abstract now specifies that QED processes regulate the *_state_* of the plasma
- Sect 5 now better communicates that our results are proof of the previously speculated picture of disk corona resembling solar corona, and
- Sect 5 specifies that our simulations with all the QED processes can reproduce *_both_* hard and soft states of radiation

Smaller point 1.

This is a smaller point, but I would not argue that it is hard to imagine how ions can get mixed in. Multiple papers have shown that there are clear mixing regions between disks, jets, jet boundaries/sheaths and coronal regions recently. I would turn the argument around and instead of “given the author's results, ions are not needed” it should be understood what the contribution of ions would be. I understand this will not be done for this work, but it is an essential part of solving the coronal problem!

Reply:

We agree. We added a caveat to Sect 5 that we assume $n_{\text{pm}} \gg n_{\text{i}}$ and noted the importance of resolving this plasma physics question with future simulations.

Smaller point 2.

A second smaller point, is the citation of thick magnetically arrested disks (papers by Liska on dynamo in thick disks, and Ripperda on thick disk MADs) in the reply to referee 1: It is completely unclear whether magnetized regions in thick disks are in any way representative for thin, luminous XRB disks.

Reply:

We agree. We specified in the text that these are simulations of *thick* disks and only *imply* a similar scenario.

Smaller point 3.

A third smaller, yet potentially important, point: It is quite unclear how the authors gets synchrotron emission. as they do not get any power law, because the effective γ_{rad} is ~ 1 . So that would make me think that the process has to be thermal “cyclotron”. However, thermal cyclotron would for real XRB coronal parameters be self absorbed. So it seems that there is some significant fine tuning to stop SSA for thermal photons? Is this potentially because the compactness is rather low, resulting in high electron temperature? This would potentially explain producing more cyclotron photons. I think this is an important point to address, and in my opinion shows that despite the first principles nature of these simulations, the results still depend strongly on assumptions and tweaking of parameters, hence my reservation to declare the two states as a solved problem.

Reply:

The high-energy tail of the cyclo-synchrotron emission naturally provides seed photons for the hard state because, at some photon energy, the Compton scattering rate exceeds the SSA rate and, therefore, photons with energies $x > x_{\text{SSA}}$ can escape the absorption. Indeed, one of the technical breakthroughs of the manuscript is the ability to accurately simulate this process---and one of the physical breakthroughs is the demonstration that the process self-consistently produces enough photons in magnetized plasmas to reproduce the Cyg X-1 hard state spectra. We illustrate the seed photon mechanism in a new section in Appendix B.

Other minor changes:

- Changed "in-falling" to "mass transfer" in the abstract because in-falling can be misunderstood as Bondi accretion.

REVIEWERS' COMMENTS

Reviewer #1 (Remarks to the Author):

The authors addressed my issues and well revised the manuscript. I reread the manuscript and I do not find any further concerns. Thus, I recommend it be published in Nature Communication in this current form.